# ATRX Loss in the Development and Prognosis of Conjunctival Melanoma

**DOI:** 10.3390/ijms241612988

**Published:** 2023-08-20

**Authors:** Jolique A. van Ipenburg, Quincy C. C. van den Bosch, Dion Paridaens, Hendrikus J. Dubbink, Emine Kiliç, Nicole Naus, Robert M. Verdijk

**Affiliations:** 1Department of Pathology, Radboud University Medical Center, Geert Grooteplein Zuid 10, 6525 GA Nijmegen, The Netherlands; 2Department of Ophthalmology, Erasmus University Medical Center, Doctor Molewaterplein 40, 3015 GD Rotterdam, The Netherlands; 3Department of Ocular Oncology, The Rotterdam Eye Hospital, Schiedamse Vest 180, 3011 BH Rotterdam, The Netherlands; 4Department of Pathology, Section Ophthalmic Pathology, Erasmus University Medical Center, Doctor Molewaterplein 40, 3015 GD Rotterdam, The Netherlands; 5Department of Pathology, Leiden University Medical Center, Albinusdreef 2, 2333 ZA Leiden, The Netherlands

**Keywords:** conjunctival melanoma, genetics, pathology

## Abstract

Metastatic disease is linked to *TERT* promoter mutations in conjunctival melanomas (CM). Both *TERT* promoter and *ATRX* mutations are associated with faulty telomere maintenance. This study aimed to determine the prognostic value of ATRX loss in conjunctival melanocytic lesions. Eighty-six conjunctival melanocytic lesions from the Rotterdam Ocular Melanoma Study group were collected. *ATRX* status and *TERT* promoter status were determined using immunohistochemical staining and molecular diagnostics, respectively. None of the nevi (*n* = 16) and primary acquired melanosis (PAM) without atypia (*n* = 6) showed ATRX loss. ATRX loss was found in 2/5 PAM with atypia without CM and in 8/59 CM. No cases with a *TERT* promoter mutation (*n* = 26) showed ATRX loss. Eight/eleven metastatic CM harbored a *TERT* promoter mutation, two other metastatic CM showed ATRX loss and one metastatic case showed no *TERT* promoter/*ATRX* alterations. In conclusion ATRX loss and *TERT* promoter mutations are only found in (pre)malignant conjunctival melanocytic lesions, with most metastatic cases harboring one of these alterations, suggesting that both alterations are associated with adverse behavior. Similar to *TERT* promoter mutations, ATRX loss may be used as a diagnostic tool in determining whether a conjunctival melanocytic lesion is prone to having an adverse course.

## 1. Introduction

Conjunctival melanomas (CM) derive from melanocytes in the conjunctiva, a transparent mucosal membrane that covers the eye and the inside of the eyelids. CM has a crude incidence of 0.39 and 0.46 per million in US and European countries, respectively. Although this disease is rare, the incidence is increasing and increases with age [1,2,3]. Recurrences of CM are common with a low survival rate for disseminated melanomas [4]. The 5- and 10-year mortality rate for CM patients is reported up to 26% and 59%, respectively [5]. This emphasizes the need for insight into factors that predict the lesion’s behavior, so that lesions with a suspected adverse course can be treated is a timely manner.

Recent developments in ocular oncology have led to a better molecular understanding of CM, which enabled improved prognostication and led to the introduction of novel therapies. CM is a mucosal melanoma with much resemblance to melanoma of the skin [6]. The majority derives from primary acquired melanosis with atypia (PAM+), but infrequently CM develops from a pre-existing nevus or de novo [7]. Similar to cutaneous melanoma, CM are often classified according to their driver mutational status, resulting in groups of *BRAF*-mutated, *NRAS*-mutated, *NF1*-mutated, and triple-wild-type (triple-WT) melanoma [8]. The triple-WT cohort appears to be heterogeneous and may alternatively carry *CTNNB1*, *c-Kit,* or *SF3B1* mutations [9]. As no relation has been reported with recurrences, metastasis, or survival, the presence of a *BRAF*, *NRAS,* or *NF1* mutation is of limited use for prognosis in CM [6,10]. We have reported that the *TERT* promoter mutation, which is present in 43% of all CM cases, is associated with reduced metastasis-free survival [11]. A *TERT* promoter mutation may be seen in all above-mentioned mutational groups [7,8].

Enhanced telomere maintenance is a key event in cellular immortality in order to escape cellular crisis, potentially occurring through activation of telomerase (*TERT*) or through alternative lengthening of telomeres (ALT), most commonly resulting from *ATRX* inactivation [12,13,14].

Because of the implications in cancer and therapy there is growing interest in *ATRX*, which is reported as a frequently mutated gene in cancer. ATRX has several domains, binding to heterochromatin with its ADD domain. ATRX forms a complex with DAXX in order to deposit histone variant H3.3 at repetitive regions, which is hypothesized to maintain chromatin structural stability. Maintenance of the repetitive regions in a heterochromatic state is essential in order to avoid aberrant transcription at telomeres. In case of ATRX loss there is decreased H3.3 deposition at telomeric ends, causing reduced chromatin compaction and increased transcription of long noncoding RNA. This leads to genome instability because of telomere dysfunction, telomeric DNA damage, and excessive telomere recombination, with the forming of secondary structures [14]. *TERT* promoter mutations and loss of function *ATRX* mutations are usually mutually exclusive, as has been observed in gliomas [15]. Approximately 25% of CM have recently been reported to exhibit inactivating mutations in *ATRX* [16]. This finding has prompted us to investigate the role of *ATRX* within the pathogenesis of conjunctival melanocytic lesions and to elucidate the prognostic value of ATRX loss in CM in addition to the known adverse effect of *TERT* promoter mutations.

## 2. Results

To determine whether ATRX loss plays a role in the development of CM, we compared the *ATRX* status in conjunctival melanocytic nevi with the *ATRX* status in CM. As most CM derive from PAM+, we compared primary acquired melanosis without atypia (PAM−) with PAM+ to determine whether ATRX loss is likely to be an early event in melanoma genesis. Subsequently, in order to determine whether ATRX loss has prognostic value in CM we evaluated *ATRX* status in CM cases with recurrent disease versus CM cases without recurrent disease as well as CM cases with metastatic disease versus CM cases without the development of metastatic disease. Moreover, because of the aforementioned prognostic relevance of *TERT* promoter mutations, we analyzed whether there might be patterns similar to *TERT* promoter mutations analyzing the *ATRX* status. In case of ATRX loss none of the atypical melanocytes showed ATRX expression in contrast to cells in the surrounding tissue, including lymphocytes and vessels, which served as an internal control in addition to the external control tissue. None of the cases showed heterogeneous ATRX expression in the (pre)malignant melanocytic lesions. In case of CM in the context of PAM + both melanocytic components showed a similar ATRX staining pattern.

### 2.1. ATRX Status in Conjunctival Melanocytic Nevi versus CM

*ATRX* status was determined in 16 nevi, with none of the cases (0%) showing ATRX loss. *ATRX* status was determined in 40 CM cases, with 9 cases (23%) showing ATRX loss (Figure 1).

The flow chart in Figure 1 concerns the analysis of the different conjunctival melanocytic lesions, depicting the results from the benign lesions (green boxes) versus the (pre)malignant lesions (red boxes), with ATRX loss only found in the (pre)malignant conjunctival melanocytic lesions. Fifteen patients with conjunctival melanoma developed recurrent disease and eleven patients with conjunctival melanoma developed metastatic disease, with most of the cases with metastatic disease harboring either a *TERT* promoter mutation or ATRX loss.

The diagnosis (conjunctival melanocytic nevus, primary acquired melanosis either with or without atypia and conjunctival melanoma) was made based on hematoxylin and eosin staining (H&E). ATRX status was (ATRX+ = no ATRX loss, ATRX- = ATRX loss, ATRX? = ATRX expression unknown) based on immunohistochemistry and *TERT* promoter status (*TERT* promoter mutation = TERT mut, no *TERT* promoter mutation = TERT wt, TERT? = *TERT* promoter status unknown) based on SNaPshot analysis and/or next generation sequencing.

### 2.2. ATRX Status in PAM− versus PAM+

*ATRX* status was determined in six cases of PAM− and five cases of PAM+ without an invasive component and seven cases of PAM+ with an invasive component. None of the cases (0%) of PAM− revealed ATRX loss. However, ATRX loss was found in two cases (40%) of PAM+ without an invasive component and five cases (71%) of PAM+ with invasive component (Figure 1).

### 2.3. ATRX Status and CM with Recurrent Disease

Twelve CM cases with known *ATRX* status developed recurrent disease, including two cases with ATRX loss. One case with recurrent disease had an unknown *ATRX* status (Figure 1).

### 2.4. ATRX Status and CM with Metastatic Disease

Nine CM cases with known *ATRX* status developed metastases, including two cases (22%) with ATRX loss. Two cases with metastatic disease had an unknown *ATRX* status (Figure 1).

### 2.5. TERT Promoter Mutation and ATRX Loss in CM

*TERT* promoter status could be determined in 58 CM cases. A *TERT* promoter mutation was present in 26 cases (45%), with none of these cases (0%) showing ATRX loss. ATRX loss was found in nine cases without a *TERT* promoter mutation (23%). A *TERT* promoter mutation was found in eight of eleven cases with metastatic disease, with no ATRX loss in six cases and unknown *ATRX* status in two cases. Two cases with metastatic disease without a *TERT* promoter mutation did show ATRX loss. Only one CM case with metastatic disease revealed neither a *TERT* promoter mutation nor ATRX loss (Figure 1).

ATRX loss could not independently be observed to be significantly associated with a shorter metastatic free survival (*p* = 0.13), while the presence of a *TERT* promoter mutation was significantly correlated with a shorter metastatic-free survival (*p* = 0.002). However, when comparing all the cases with presumed enhanced telomere lengthening changes as a result of *TERT* or *ATRX* aberrations versus cases without these aberrations, we found that the presumed telomere affected group was associated with a significantly shorter metastasis-free survival (*p* = 0.01) (Figure 2). Due to the low number of cases, it was not possible to adjust for clinicohistopathological features, which we previously reported to be associated with a shorter metastasis-free survival [11]. Fifteen CM cases developed recurrent disease, including four cases with a *TERT* promoter mutation. Of the eleven cases without a *TERT* promoter mutation two cases showed ATRX loss. No significant correlation with recurrent disease was observed (*p* > 0.05).

## 3. Discussion

In non-neoplastic tissue the repetitive DNA sequences at the terminal ends of the chromosome progressively shorten during each cell division. In cases of uncontrolled proliferation several telomere maintenance mechanisms (TMM) are described. Eighty to ninety percent of all tumors reactivate telomerase, an enzyme consisting of multiple proteins, adding TTAGGG at the terminal end of the chromosome when the telomeres are getting too short. In contrast, in ten to twenty percent of all tumors telomere elongation is achieved via ALT. This second type of TMM does not rely on a single enzyme and results from newfound complementary sequences, which serve as elongation templates for the invading strand, including the homologous chromosome, an unrelated chromosome or extrachromosomal telomeric repeat [17]. ALT activation is described in sarcoma, astrocytoma, neuroblastoma, and carcinoma [18] cases as well as CM [16] and is related to functional loss of ATRX and/or histone H3.3 chaperone DAXX. In mucosal-melanoma-inactivating *ATRX* mutations can be revealed using immunohistochemistry [19]. In the current study ATRX loss was only found in (pre)malignant lesions and not in benign lesions. This finding of ATRX loss in PAM+ as well as in CM is congruent with findings in other studies, that suggested that *TERT* promoter mutations [20] and ATRX loss may be early events in CM progression [16]. Moreover, the presence of a *TERT* promoter mutation was evaluated, with the usually mutually exclusive presence of *TERT* promoter mutations and *ATRX* mutations described in gliomas [21] as well as cutaneous melanomas [22]. This suggests that the functional effects of the different telomere maintenance mechanisms are similar. Consistent with these studies and other mucosal melanoma studies [13,16,19], our current study indicates that ATRX loss and *TERT* promoter mutations are also mutually exclusive in CM. If enhancement of telomere maintenance is associated to an adverse clinical course for conjunctival melanocytic lesions, CM associated with metastasis but without detected *TERT* or *ATRX* alterations may harbor alterations in other components of the machinery involved in telomere maintenance, including *POT1* [21]. Another less frequently occurring explanation may be that in cases without a *TERT* mutation and ATRX loss, a loss of function *ATRX* mutation is missed due to the limitations of the detection method [18].

Conjunctival melanoma is a mucosal melanoma and, based on the location, it is also an ocular melanoma. The genetic make-up partly overlaps with other ocular melanomas and mucosal melanomas originating from other sites. In addition, there are many similarities with the genetic make-up of cutaneous melanomas. Depending on the location, cutaneous melanomas are frequently exposed to ultraviolet radiation (UV) as are most CM. Melanomas from both sites frequently harbor the UV-signature (C > T nucleotide transition) [20]. Because of this signature it is likely that UV exposure (partly) plays a role in pathogenesis in at least a subset of CM. Unlike *TERT* promoter mutations, the relationship between UV exposure and ALT is not as well comprehended. However, there is a suggested connection between UV exposure and telomere changes, supported by the discovery of shorter telomeres in sun-exposed skin compared to nonexposed skin. It has been hypothesized that nucleotide excision repair restores telomeres by removing DNA photoproducts that are formed at telomeres after UV exposure. Lower levels of photoproducts at telomeres are formed due to shelterin protection [23]. A mutated shelterin complex, e.g., by *POT1* mutations, might activate a noncanonical mechanism of ALT due to shelterin dysfunction [24]. If there is a relation between an *ATRX* mutation and UV exposure, maybe by interaction with the noncanonical mechanism of ALT, this still remains to be elucidated. A subset of CM reveals ATRX loss. *ATRX* mutations are also described in other mucosal melanomas, including melanomas from gynecologic sites [19], the anorectal region, and the oral region [13,25]. Similar to *TERT* promoter mutations [26], ATRX loss is very rare in uveal melanoma [27], which is another ocular melanoma [20]. The finding of *ATRX* mutations in melanoma originating from locations without sun exposure and the association with non-sun-exposed conjunctiva [16] suggest that a (direct) relation between UV and *ATRX* is less likely.

In the current study the presence of a *TERT* promoter mutation was strongly associated with a shorter metastasis-free survival, as was also described in other studies [11,20]. In publicly available datasets *ATRX* aberrations (including deep deletions or mutations) are reported in melanoma in general, with data suggesting a more favorable course for *ATRX*-unaltered cancer (according to cBioPortal database and Human Protein Atlas, accessed on 1 August 2023), although the data for *ATRX*-altered cases are limited, with a lack of specific information concerning the melanoma arising from different locations, especially mucosal melanoma. However, this information supports our hypothesis and certainly warrants further exploration. In an earlier study, the presence of an *ATRX* mutation has been suggested to be associated with a beneficial clinical course compared to *ATRX* wildtype CM, although a significant association with overall survival was not found [16]. The limitations of this study, as acknowledged by the authors, include a restricted median follow-up period, which may introduce biases to the reported findings [16]. Due to the counterintuitive nature of the findings in the previous study, given the presumed underlying mechanism, we performed the current study. We found a similar pattern for ATRX loss and *TERT* promoter mutations. Our hypothesis was supported by the finding of a significantly shorter metastasis-free survival in the presumed telomere affected group versus the group without *TERT* promoter or *ATRX* aberrations. The lack of a significant direct correlation between ATRX loss and metastasis-free survival is likely attributed to the lower occurrence of ATRX loss (23%) compared to *TERT* promoter mutations (40%) and the subsequent inclusion of a smaller number of cases with ATRX loss in the analysis. This supposition is further supported by the finding that the majority of CM with metastatic disease revealed either ATRX loss or a *TERT* promoter mutation.

## 4. Materials and Methods

### 4.1. Material Selection

We collected 16 conjunctival melanocytic nevi, 6 PAM without atypia (PAM−), 5 PAM+ without an associated CM, 7 PAM+ with an associated CM, and 59 CM, diagnosed between 1987 and 2016 at the Erasmus MC University Medical Center (Rotterdam, The Netherlands) and The Rotterdam Eye Hospital (Rotterdam, The Netherlands). Clinicopathological characteristics of the CM cases are described in Table 1, with a median follow-up time of 35 months (range 0–257 months). Eleven CM (19%) showed metastatic disease (median 15 months (range 2–49 months)) and 13 CM showed recurrent disease (median 23 months (range 5–157 months)) (Table 1).

Ethics Committee approval was obtained by the Medical Ethics Committee, Erasmus University Medical Center, Rotterdam, The Netherlands (4 October 2018), and was registered with reference 67865. The study was performed according to the tenets of the Declaration of Helsinki. Data regarding gender, age at the time of diagnosis, location, tumor thickness, the origin of the lesion, and information on the development of recurrences and metastasis were collected from the patient records and information was obtained from the pathology reports and the nationwide pathology network and registry system (Pathologisch-Anatomisch Landelijk Geautomatiseerd Archief).

Recurrence was defined as described earlier [11] with recurrence-free survival and metastasis-free survival defined as the time from the primary treatment to the date of recurrent disease and metastatic disease, respectively, or last date of follow-up.

*TERT* promoter status was determined in cases of CM using a sufficient DNA quantity and quality, to comply with the manufacturer’s instructions. *TERT* promoter mutation status was determined using SNaPshot analysis and/or next generation sequencing, as described earlier [20,26]. We did not find any discrepancies in *TERT* promoter status in cases with sufficient DNA quantity and quality to validate *TERT* promoter status using both techniques.

ATRX expression was determined using immunohistochemistry (IHC), with staining of the samples using an automated IHC staining system (Ventana Benchmark ULTRA, Ventana Medical System Inc., Tucson, AZ, USA) in an ISO 15189:2012 certified laboratory. The tissue sections were incubated for 32 min at 37 °C with a rabbit–anti-human *ATRX* polyclonal IgG antibody (Atlas Antibodies, Bromma, Sweden, Product number HPA001906, dilution 1:150). Human tonsil tissue was used as control tissue in addition to internal control tissue. The hematoxylin and eosin (H&E) staining and IHC staining was evaluated by an ophthalmic pathologist (RVE), with inactivation of *ATRX* reflected as no expression of ATRX (Figure 3), hereafter referred to as ATRX loss. For 19 cases it was not possible to reliably assess *ATRX* status because of ambiguous staining or there was no material available for further analysis.

### 4.2. Statistical Analysis

Statistical Package for Social Science (IBM SPSS Statistics, version 25) was used for data analysis, with differences between categorical data calculated using either the Pearson’s χ^2^ test or the Fisher’s exact test. Kaplan Meier survival curves were made using log-rank statistics. *p* < 0.05 is considered statistically significant.

## 5. Conclusions

Conjunctival melanocytic lesions may be difficult to characterize adequately based solely on clinicopathological characteristics. One of the conclusions drawn from this study is that ATRX loss can be helpful in distinguishing between benign and (pre)malignant conjunctival melanocytic lesions. A correlation was observed between the presence of a *TERT* promoter mutation and decreased metastasis-free survival. The findings related to CM with ATRX loss indicate a similar trend. Since most CMs with metastatic disease show either ATRX loss or a *TERT* promoter mutation, both alterations are suggestive of an association with adverse clinical behavior. Our study shows that the presence of a *TERT* promoter mutation in CM is strongly associated with a shorter metastasis-free survival, and is indicative that this may also hold for CM with ATRX loss. The major implications of these findings warrant further research using larger multicenter cohort studies. For this reason, an international collaborative study on the prognostics of *TERT* promoter mutation and ATRX loss has been initiated by the European section of the Ophthalmic Oncology Group, an independent scientific group devoted to clinical ophthalmic oncology and related basic science research.

## Figures and Tables

**Figure 1 ijms-24-12988-f001:**
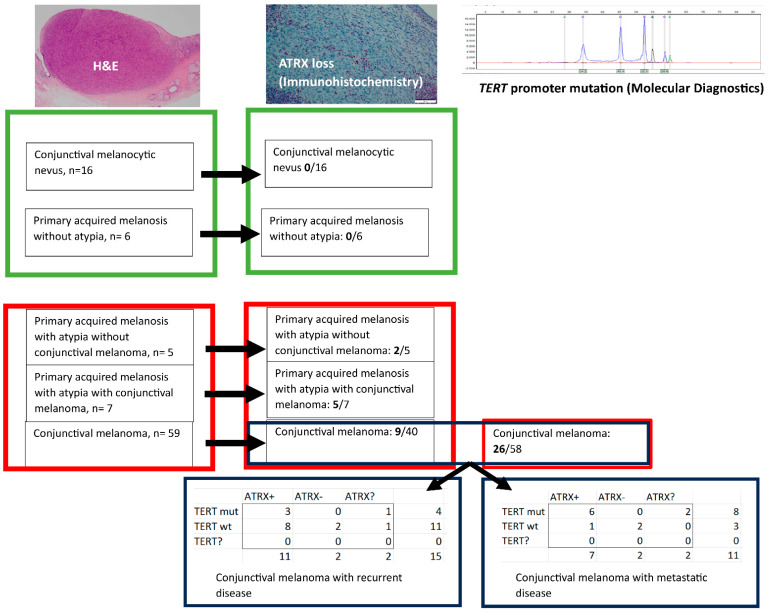
*TERT* promoter and *ATRX* status in conjunctival melanocytic lesions. These images are meant to highlight the methods that are used to obtain the data (H&E staining and immunohistochemistry instead of other (diagnostic) methods; Microscope's magnification or scale bar is not of relevance for understanding this figure. Exact data concerning these images can be provided upon request.

**Figure 2 ijms-24-12988-f002:**
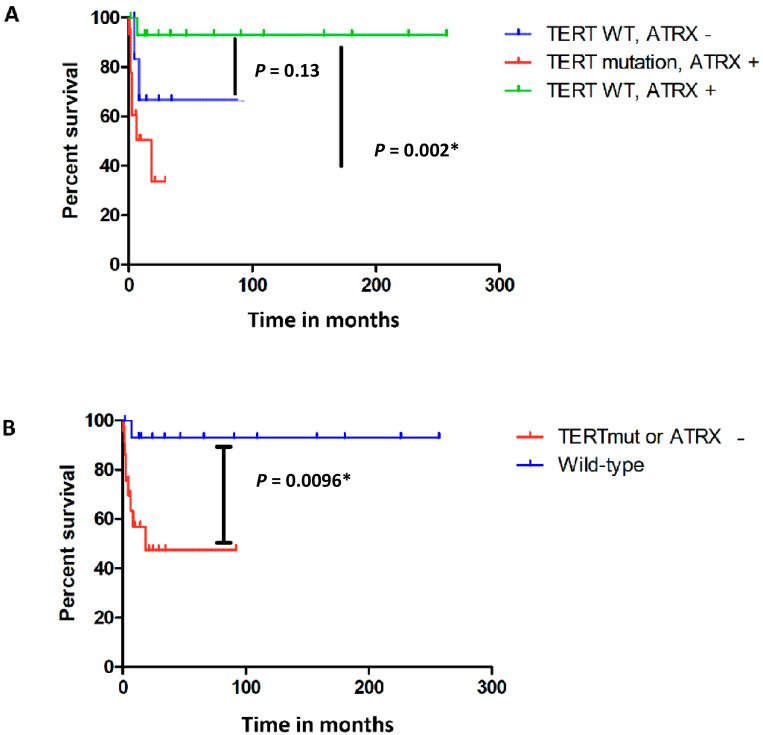
Kaplan Meier *ATRX* and *TERT* aberrations in conjunctival melanoma cases with metastatic disease. The Kaplan Meier survival estimate for the time to metastasis of presumed telomere affected conjunctival melanoma (CM) (cases with a *TERT* promoter mutation (“TERTmut”) and cases with ATRX loss (“ATRX−”) shown separately) shows that CM cases harboring a *TERT* promoter mutation (depicted in red) have a significantly shorter metastasis-free survival compared to CM cases without a *TERT* promoter mutation or ATRX loss, the presumed telomere unaffected CM (depicted in green). This pattern for CM with ATRX loss is similar to CM cases with *TERT* promoter mutations, suggesting that ATRX loss is also associated with a shorter metastasis-free survival (**A**). Although this finding concerning the cases with ATRX loss is not proved to be significant (*p*-value 0.13) (**A**), the hypothesis is supported by the Kaplan Meier survival estimate in (**B**), showing that the presumed telomere-affected CM (depicted in red) have a significant shorter metastasis-free survival compared to the CM without *TERT* promoter and *ATRX* aberrations (depicted in blue). * = statistically significant.

**Figure 3 ijms-24-12988-f003:**
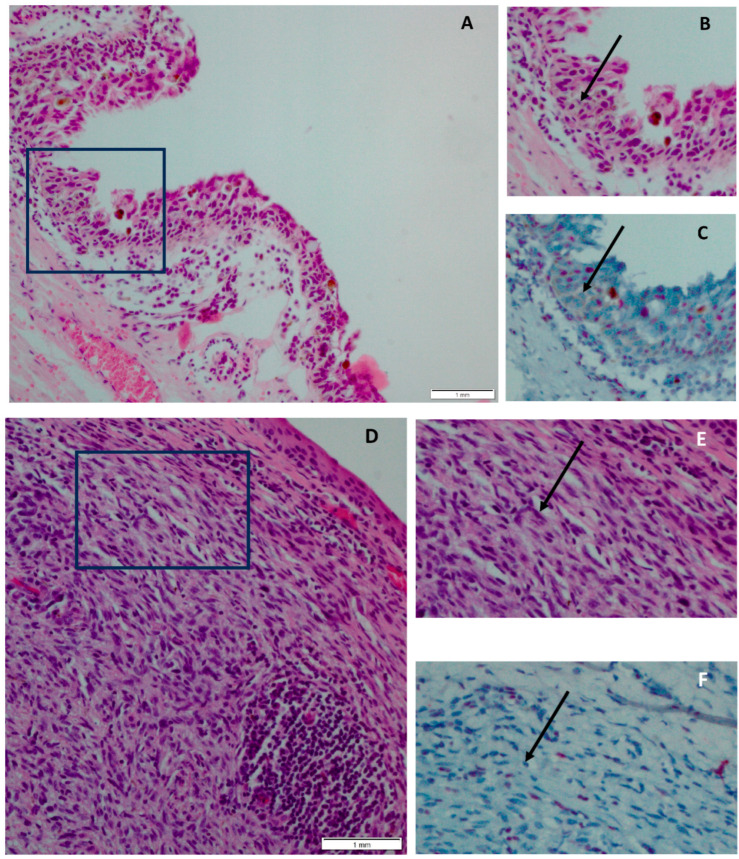
Determination of *ATRX* inactivation using immunohistochemical staining as determined in (pre) malignant conjunctival melanocytic lesions. H&E staining within the box in (**A**) primary acquired melanosis with atypia (PAM+), magnification 200× (scale bar 1 mm), with the PAM+ highlighted in (**B**), shows ATRX loss in the atypical melanocytes in (**C**), with the atypical melanocytes depicted with an arrow. *ATRX* status was evaluated using immunohistochemistry, with the red stained cells representing pre-existing tissue, including normal epithelium and fibroblasts, serving as an internal control. H&E staining within the box in (**D**) conjunctival melanoma, magnification 200× (scale bar 1 mm), with the stromal part with atypical melanocytes (conjunctival melanoma) highlighted in (**E**), shows ATRX loss concerning the atypical melanocytes in (**F**), with the atypical melanocytes depicted with an arrow.

**Table 1 ijms-24-12988-t001:** Clinicohistopathological characteristics of the conjunctival melanomas.

	Age at Time of Diagnosis (y)	Gender	Recurrent Disease, Months after Diagnosis	Metastasis, Months after Diagnosis	Breslow (mm)	Ulceration	Presence of Mitotic Figure (s)	Presence of Epithelioid Cells	pT Status	Origin	Diameter (mm)	Bulbar Involvement	Follow-Up (Months after Diagnosis)
1	43	M	20	47	UK	Yes	Yes	Mixed	pT2b	UK	7	UK	68
2	62	M	No	No	UK	Yes	Yes	Mixed	UK	UK	17	UK	4
3	44	M	No	38	1.7	Yes	Yes	Mixed	pT1a	Nevus	21 *	UK	38
4	65	F	55	No	3.46	Yes	Yes	Mixed	pT2b	PAM+	7 *	UK	257
5	60	F	No	No	0.4	No	Yes	Mixed	UK	De novo	6 *	UK	1
6	68	F	No	8	7	Yes	Yes	Mixed	UK	PAM+	18	UK	54
7	84	M	No	No	2.6 *	Yes	Yes	Mixed	pT1b	De novo	6 *	UK	1
8	65	M	No	No	3	Yes	Yes	Spindle cells	pT1b	PAM+	6	UK	6
9	64	M	No	No	1.55 *	No	Yes	Mixed	pT1a	Nevus	6	UK	29
10	51	M	8	9	0.96 *	No	Yes	Spindle cells	UK	De novo	12 *	UK	22
11	49	M	No	49	2.3 *	Yes	Yes	Mixed	UK	PAM+	3	UK	86
12	84	M	23	No	2.3 *	UK	UK	Epithelioid	UK	PAM+	UK	UK	35
13	73	F	157	No	3	UK	Yes	Spindle cells	UK	PAM+	UK	UK	158
14	53	M	No	14	3.69	Yes	Yes	Epithelioid	UK	Nevus	7	UK	36
15	64	M	81	No	1.1	No	Yes	Spindle cells	pT1a	PAM+	UK	UK	109
16	49	F	No	No	0.3	No	No	Mixed	pT1a	PAM+	3	UK	166
17	79	F	No	No	UK	UK	UK	UK	UK	PAM+	UK	UK	21
18	66	M	No	No	2	Yes	No	Mixed	pT1a	De novo	4	UK	9
19	83	F	87	No	5.1	No	Yes	Mixed	pT2b	PAM+	6 *	UK	90
20	46	F	No	2	7 *	Yes	Yes	Mixed	UK	PAM+	11	UK	69
21	65	M	11	15	1.9 *	Yes	Yes	Mixed	UK	De novo	4.2	UK	58
22	63	M	No	No	UK	UK	UK	UK	UK	PAM+	5	UK	24
23	40	M	No	No	0.3	No	No	Spindle cells	pT1a	PAM+	1.3	Yes	3
24	55	F	No	No	0.35	No	No	Spindle cells	pT1a	PAM+	10	Yes	187
25	75	F	5	No	0.37	No	No	Spindle cells	pT1a	PAM+	2	Yes	226
26	71	F	No	No	0.5	No	No	Spindle cells	UK	PAM+	UK	UK	1
27	85	M	No	No	0.5	Yes	Yes	Spindle cells	pT1a	De novo	UK	Yes	0
28	54	M	No	No	0.5	No	No	UK	pT1a	PAM+	UK	Yes	142
29	72	M	No	No	0.6	No	No	Mixed	pT1a	PAM+	9 *	Yes	92
30	56	F	No	No	0.62	No	No	Mixed	UK	Nevus	4.5 *	UK	185
31	68	M	No	No	0.7	Yes	No	Spindle cells	pT1a	PAM+	UK	Yes	112
32	43	M	No	No	0.78	No	No	Spindle cells	pT2a	De novo	2.5	No	69
33	52	M	No	No	0.9	No	No	Epithelioid	UK	PAM+	UK	UK	15
34	84	M	No	No	0.9	UK	UK	Mixed	pT1a	PAM+	UK	Yes	24
35	73	M	No	No	1.3	Yes	Yes	Mixed	pT1a	De novo	UK	Yes	3
36	67	F	No	No	1	Yes	Yes	Epithelioid	pT1a	PAM+	6 *	Yes	2
37	16	F	No	18	1.04	Yes	Yes	Mixed	pT1a	Nevus	0.5	Yes	180
38	57	M	No	No	1.05	No	Yes	Epithelioid	pT2a	UK	UK	No	2
39	63	F	No	No	1.1	No	No	UK	pT1a	Nevus	3	Yes	54
40	56	M	No	No	1.3	No	Yes	Mixed	pT1a	PAM+	4.5 *	Yes	0
41	76	F	No	No	2.3	Yes	Yes	Mixed	pT1b	PAM+	17 *	Yes	2
42	51	F	No	No	3	Yes	Yes	Spindle cells	pT2b	De novo	10	No	95
43	89	M	No	No	7.7 *	Yes	Yes	Mixed	pT3a	PAM+	14	No	13
44	41	M	No	No	1 *	No	Yes	Spindle cells	UK	Nevus	6	Yes	14
45	69	F	8	No	1.1 *	No	No	Mixed	UK	PAM+	UK	Yes	34
46	66	M	No	No	1.1 *	No	Yes	Spindle cells	UK	PAM+	4 *	yes	81
47	70	F	No	No	1.2 *	No	No	Mixed	UK	UK	4	Yes	13
48	71	M	No	No	2.7 *	Yes	Yes	Mixed	pT1b	De novo	6	Yes	179
49	57	M	No	No	2.5 *	Yes	UK	Mixed	pT1b	PAM+	6	Yes	15
50	79	F	No	No	2.3	UK	UK	UK	UK	PAM+	0.7	Yes	0
51	73	M	No	No	6.2	UK	UK	UK	UK	PAM+	1.2	No	22
52	83	F	No	No	3	UK	UK	UK	UK	Nevus	1	Yes	0
53	80	F	29	No	0.2	UK	UK	UK	UK	PAM+	0.8	Yes	73
54	74	F	No	45	UK	UK	UK	UK	UK	PAM+	UK	Yes	49
55	37	F	No	7	4	UK	UK	UK	UK	UK	UK	No (though the caruncle is involved)	66
56	56	M	7	No	3.5	UK	UK	UK	UK	PAM+	UK	Yes	47
57	58	M	42	No	3	UK	UK	UK	UK	PAM+	0.6	Yes	174
58	60	M	No	No	1.1	UK	UK	UK	UK	UK	0.6	No	5
59	55	M	No	No	2.5	UK	UK	UK	UK	UK	0.7	UK	4

Clinical and histopathological characteristics of the cases with conjunctival melanoma. PAM+: primary acquired melanosis with atypia; UK: unknown; * these values are the minimum values as in these cases it was not possible to determine the exact value since the lesion was not completely excised and/or not all slides were available for review. Mixed: presence of both spindle cells and epithelioid cells.

## Data Availability

Data can be provided upon request.

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
