# Peer review of "ATRX Loss in the Development and Prognosis of Conjunctival Melanoma"

_ijms, 2023, doi:10.3390/ijms241612988_

Round 1
Reviewer 1 Report
Sir, I have reviewed the manuscript "ATRX loss in the development and prognosis of conjunctival melanoma: submitted by Jolique Van Ipenburg and co-workers to IJMS recently. Ocular melanoma is a relatively rare disease with significant mortality. Our therapeutical options here are somewhat limited compared to more frequent cutaneous melanomas. The authors aimed to determine whether the presence of a TERT promoter mutation in conjunctival melanoma is somehow associated with a worse prognosis (in oncological terms of progression-free survival). Major point: The study is based on a somewhat limited number of cases. However, given by the relative rarity of the disease, it is still a considerable number. The Introduction is very brief. However, I believe that ATRX should be (at least) briefly introduced here in a separate paragraph for the benefit of interested readers. Relevant to IHC findings: ATRX expression was determined using immunohistochemistry (IHC). However, I am sorry to say that the description of findings in individual categories (2.1-2.4) is insufficient and has a relatively low informative value. The authors stated that (quote, lines 229-230) "with inactivation of ATRX reflected as no expression of ATRX (Fig. 2), hereafter referred to as ATRX loss. Were there any cases showing heterogeneity in ATRX staining pattern? This aspect could be greatly interesting to the readers as it can suggest some intratumoral evolution. I believe that pathologists could provide more structured information. Also, Figure 2 is very uninformative in this aspect. It would be greatly appreciated to consider better image selection (maybe higher magnification). Notably, the authors are completely silent regarding negative controls used in their IHC study (I hope they do not refer to negative controls as "in addition to internal control tissue"). In IHC-based research, this is not a formal issue. In the Discussion (lines 133-134), the authors quote: "Inactivating ATRX mutations can be reliably detected with immunohistochemistry in mucosal melanoma". I would gently suggest being more stringent here and improving the wording. Such a straightforward conclusion can be potentially misleading. Relevant to the material - lines 220-221 - the quality of material submitted for mutational analysis is poorly described. Minor points: Relevant to Table 2. Clinicohistopathological characteristics of the conjunctival melanomas: the formatting (portrait) is unsuitable, due to poor legibility of the table. Please, consider landscape orientation (and potentially a better graphical style). Relevant to Figure 1, panel B: probably missing "p= ...%" ...as above in panel A (plus highlight the statistical significance by an asterisk, if significant). Please, improve the wording of the legend. Relevant to Figure 2: please, consider better graphical quality. Also, the marking of the scalebar is not legible. To draw a conclusion, I believe that the topic is worthy of attention and could be potentially interesting. The authors summoned on a limited number of available cases some evidence. The authors are also right that their findings warrant further research using larger multicenter cohort studies. I believe that the Suggested improvements could be made in nearest future. I am also keen to review the resubmitted manuscript again.Author Response
Please see the attachment.

Reviewer 2 Report
The author aimed to determine the prognostic value of ATRX loss in conjunctival melanocytic lesions. They revealed that TRX loss and TERT promoter mutations are only found in (pre-)malignant conjunctival melanocytic lesions, with most metastatic cases harboring one of these alterations, suggesting that both alterations are associated with adverse behavior. Similar to TERT promoter mutations ATRX loss may be used as a diagnostic tool in determining whether a conjunctival melanocytic lesion is prone to have an adverse course. Albeit, I consider these findings to provide new insight into melanoma-related fields, I still have some suggestions.
1, Most figures and tables are highly professional; however, the authors should guide the readers to the meaning of the images appropriately; otherwise, it will likely cause misunderstandings. Therefore, I suggest the author consider revising these figure and tables legends again.
2, The author investigated the role of ATRX within the pathogenesis of conjunctival melanocytic lesions and to elucidate the prognostic value of ATRX loss in CM in addition to the known adverse effect of TERT promoter mutations. However, it would be much better if the authors could provide some Workflow or Scheme for this research, I suggest that they can take a look at the recent paper in MDPI (PMID: 35563422, 36677020)
3, In Figure 2, the author presented the determination of ATRX inactivation using immunohistochemical staining as determined in (pre-) malignant conjunctival melanocytic lesions. However, I suggest the authors can validate their data via Proteinatlas or cBioportal, and discuss these methodologies and related literature in the manuscript (PMID: 17008526, 22588877).
4, There are few typo issues for the authors to pay attention to; please also unify the writing of scientific terms. “Italic, capital”? Please also double-check superscripts and subscripts for the whole manuscript.
5, Most references are out of date, the author needs to discuss the recent paper as well as the analysis methods in this manuscript. Meanwhile, the introduction part need rewrite and present the purpose of the investigation and cite pertinent literature.
Minor editing of English language required
Round 2
Reviewer 1 Report
Sir,
I have reviewed the new version of the manuscript of Dr. van Ipenburg and co-workers. I have studied all their point-by-point answers provided in the rebuttal letter with utmost care. They have honestly followed all my critical points raised in the initial review. They have provided all necessary modifications to the text and also to the graphical part of their manuscript. As it is in this version, the submitted work is ready for me, and I am happy to support the publication of their findings. I believe it can easily find interested readers.